# Contraceptive confidence and timing of first birth in Moldova: an event history analysis of retrospective data

Mark Lyons-Amos,[1] Sabu S Padmadas,[2] Gabriele B Durrant[3]

▶ Prepublication history and additional material is available. To view please visit the journal (http://dx.doi.org/10.1136/bmjopen-2014-004834).

[1]Department of Quantitative Social Science, Institute of Education, London, UK
[2]Faculty of Social & Human Sciences, Centre for Global Health, Population, Poverty & Policy, University of Southampton, Southampton, UK
[3]Faculty of Social & Human Sciences, Southampton Statistical Sciences Research Institute, University of Southampton, Southampton, UK

**Correspondence to**
Dr Mark Lyons-Amos;
mark.lyonsamos@gmail.com

## ABSTRACT

**Objectives:** To test the contraceptive confidence hypothesis in a modern context. The hypothesis is that women using effective or modern contraceptive methods have increased contraceptive confidence and hence a shorter interval between marriage and first birth than users of ineffective or traditional methods. We extend the hypothesis to incorporate the role of abortion, arguing that it acts as a substitute for contraception in the study context.

**Setting:** Moldova, a country in South-East Europe. Moldova exhibits high use of traditional contraceptive methods and abortion compared with other European countries.

**Participants:** Data are from a secondary analysis of the 2005 Moldovan Demographic and Health Survey, a nationally representative sample survey. 5377 unmarried women were selected.

**Primary and secondary outcome measures:** The outcome measure was the interval between marriage and first birth. This was modelled using a piecewise-constant hazard regression, with abortion and contraceptive method types as primary variables along with relevant sociodemographic controls.

**Results:** Women with high contraceptive confidence (modern method users) have a higher cumulative hazard of first birth 36 months following marriage (0.88 (0.87 to 0.89)) compared with women with low contraceptive confidence (traditional method users, cumulative hazard: 0.85 (0.84 to 0.85)). This is consistent with the contraceptive confidence hypothesis. There is a higher cumulative hazard of first birth among women with low (0.80 (0.79 to 0.80)) and moderate abortion propensities (0.76 (0.75 to 0.77)) than women with no abortion propensity (0.73 (0.72 to 0.74)) 24 months after marriage.

**Conclusions:** Effective contraceptive use tends to increase contraceptive confidence and is associated with a shorter interval between marriage and first birth. Increased use of abortion also tends to increase contraceptive confidence and shorten birth duration, although this effect is non-linear—women with a very high use of abortion tend to have lengthy intervals between marriage and first birth.

## Strengths and limitations of this study

- The study uses a nationally representative survey.
- Use of regression analysis disentangles the net effects of related contraceptive and abortion behaviour.
- Use of retrospective data necessitates reliance on proxy measures.

unprecedented decline in fertility with a total fertility rate at or below 1.3 children/woman.[1][2] Economic uncertainty and high male out-migration partly explain the stagnant low fertility trends, although recent data show gradual recovery of fertility rates in some countries.[1][3] Many women in Moldova tend to control their fertility by using traditional contraceptive methods or induced abortions since modern method access is limited.[4–6] This research focuses on Moldova where abortions are widely practised and often accepted as a birth control method.

The dynamics of contraceptive use including discontinuation rates, switching and method efficacy is widely acknowledged in demographic research.[7–10] However, the confidence which women have in their contraceptive method and the effect it has on fertility behaviour is under-researched. Contraceptive confidence is a hypothesis which explains the timing of childbearing resulting from the perceived efficacy of contraceptive methods, but there is little modern literature[11] and much work examines older demographic data.[12][13] Theoretically, women who use less-effective contraceptive methods (traditional methods) have low contraceptive confidence, since their method is likely to fail. These women tend to space their fertility as a means to limit their intended family size.[12] In contrast, women who use effective (modern) contraceptives have a high degree of confidence that these methods will not fail. This has

## INTRODUCTION

Over the past two decades, many countries in Eastern Europe have experienced an

prompted women to compress their fertility into shorter periods.[14] [15]

While previous studies have addressed second and later birth intervals, the demographic landscape of Europe has undergone unprecedented changes in recent decades driven mostly by changes in the relationship between partnership formation—particularly marriage—and childbearing.[1] [16] These trends are gradually emerging in Moldova signalling the features of a second demographic transition[17] exemplified mostly in terms of low fertility rates (Moldovan fertility fell below 1.3 in 1999, and has bucked trends in recovering fertility seen in other countries in the region with persistent lowest-low fertility[3]) accompanied by a modest decrease in marriage rates and increasing non-marital childbearing.[18] That said, some of the trends of the second demographic transition are not present (the average age at first marriage is still low at 21, authors' calculations from the Moldovan Demographic and Health Survey (MDHS) data set). Additionally, there have been a number of other explanations for changing fertility across Eastern Europe (eg, more orthodox economic factors), and the cause is still debated among demographers and dependent on context.[19] [20] Therefore, any analysis exploring fertility behaviour should account for marriage cohort as an important control variable, albeit not one that can offer a complete explanation of observed trends.

We note that the pattern of union formation is an exceptionally complex demographic process.[11] As well as the control variables we are able to include, there will typically be significant variations in behaviour that are important but not captured by the type of representative sample survey we employ. Therefore, while we are able to describe part of the effects on first birth, this analysis should not be interpreted as a complete picture.

In Moldova, traditional methods are still widely used: about 26% of the contraceptive methods used in Moldova are traditional.[20] [21] This is considerably higher than that observed even in other former-Socialist countries (Latvia 8.7%, Hungary 9.0% and Bulgaria 15.7%).[4] Moldova therefore lends itself to examining the differential effects of contraceptive confidence on reproductive behaviour. Another characteristic of fertility control behaviour in Moldova is the widespread use of abortion: 46% of ever-sexually active women reported having had at least one abortion and about 40% of these women have had two or more abortions.[21] The widespread use of traditional contraceptives and method failure are associated with multiple abortions.[21] [22] This paper analyses the effect of contraceptive confidence on the timing of first birth, using data from the first-ever Demographic and Health Survey conducted in Moldova in 2005. The underlying research question is: To what extent does contraceptive confidence influence women's fertility behaviour and the timing of first birth? Examining first birth is an extension of the contraceptive confidence hypothesis not previously explored in demographic literature.[12] The analysis also extends the

contraceptive confidence hypothesis to capture the effect of abortion, often regarded as a method substitute to ineffective contraceptive use.[22–26] The proposition is that women who use abortion either in the event of a method failure or as a substitute for modern contraception have increased contraceptive confidence and these women are more inclined to have a first birth sooner than their counterparts.

The analysis considers marriage cohorts to capture changes in first birth rate as well as to ascertain the possible effects of exogenous economic uncertainty and poverty in delaying first birth. Other analyses[27] have observed dramatic influences of macrolevel economic factors on cohort-order specific fertility rates due to declining macroeconomic indicators and similar effects are likely in Moldova.[21] The progression to first birth was rapid among young couples during the Socialist era necessitated as a precondition to obtain housing.[23] [26] [27] Although marriage remains nearly universal, fertility behaviour postmarriage has undergone considerable changes including a delaying trend in childbearing typical of broader westernisation and modernisation processes underway in Moldova,[23] or wider demographic trends such as the Second Demographic Transition.[17]

## DATA AND METHOD
### Data and analysis sample
Data for this study are drawn from the birth history schedule of the 2005 MDHS. Details of the MDHS, including the sample design and questionnaire, are available elsewhere.[21] The date of marriage is considered as the start date of exposure since information on the date of first intercourse exhibits a much greater degree of missing data and recall error. While cohabitation has become more significant as a partnership form in Eastern Europe, the proportion of women who are in persistent non-marital cohabitation in Moldova is still below 6% (MDHS 2005) and marriage is still the socially normative relationship form for childbearing.[23]

From the original MDHS sample of 7440 women, 1884 women were excluded since they were never married and 74% of these reported having never had sex. In addition, 179 women who had premarital births (2.4%) were excluded since the terminal event (first birth) preceded the start event (marriage). The final selected sample considers 5377 married women. About 15% of births occurred within 9 months of marriage—indicative of premarital conception. The MDHS also includes detailed information of abortion histories including the number and timing of each abortion.

### Method
The analysis uses a piecewise-constant hazard model. The dependent variable is the timing of first birth (terminal event) since first marriage (start event), recorded in months and expressed as $y_i(t)$, a binary random variable for each time piece following marriage, where:

$y_i(t) = 0$ if woman i has a birth at time t, and $y_i(t) = 0$ if woman i does not experience birth at t. The hazard of a first birth is defined as $\lambda_i(t) = \Pr(y_i(t) = 1 | y_i(t-1) = 0)$, which is the hazard of experiencing a first birth in piece t conditional on not having experienced first birth in piece t−1. The effect of covariates on $\lambda_i(t)$ is estimated by the regression model described in Equation 1.

$$\ln\left[\frac{\lambda_i(t)}{1 - \lambda_i(t)}\right] = \alpha(t) + \beta(t)x(t)_i \qquad (1)$$

In equation 1, $\lambda_i(t)$ is the hazard of a first birth at time t for woman i, $\alpha(t)$ is a vector of dummy variables capturing the duration since marriage (in categories of months), $\beta(t)$ is a vector of time-dependent coefficients and $x(t)_i$ a vector of explanatory variables for women i. Where variables are time constant $x(t = 1)_i = x(t = 2)_i = x(t = T)_i$ and $\beta(t = 1) = \beta(t = 2) = \beta(t = T)$.

A piecewise-constant hazard model uses a simplified data structure compared with a standard discrete-time model, as the duration variable is collapsed into intervals, across which the hazard of a birth is assumed to be constant. The advantage of this is that the baseline hazard distribution $\alpha(t)$ and parameter estimates (β) are still unbiased, and the data set required for the analysis is considerably reduced when compared with the standard discrete-time model.[9] We test for time-dependent effects of the coefficients in the model by testing the significance of interaction terms between a(t) and β. Where interactions improve the model fit, this is considered as evidence of time dependency. To examine the possible interaction between abortion and ineffective method use, an interaction between the variable capturing contraceptive confidence and propensity to use abortion is specified in addition to the main effects. Since the final model includes many interactions, the interpretation of the coefficient directly is extremely difficult. Therefore, we use the model to generate survival curves and cumulative hazards, which are presented for interpretation.

## Explanatory variables

The main interest in the analysis of the first birth interval is the degree of contraceptive confidence. As noted by Ní Bhrolcháin,[12] the perfect measure of contraceptive confidence would include information on contraceptive tastes and preferences collected contemporaneously with use. Ní Bhrolcháin[12 14 15] argues that in the absence of this information the best available proxy is the most recent contraceptive method. We note that women may have changed their contraceptive method since their first birth, and hence our estimated contraceptive confidence may not necessarily correspond to the method used preceding the first birth. While the MDHS does include data on current contraceptive use in the contraceptive calendar, these data pertain to the 5 years prior to the survey. Using these data is not considered feasible since (1) there is only a small number

of first births in that interval (fewer than 140) and (2) the recency of the births would severely constrain our ability to make inference particularly for older marriage cohorts. About 57% of sexually active women in the MDHS have reported not switching their contraceptive method within the past 5 years. This is an important observation which validates the assumption that women in Moldova are unlikely to switch their contraceptive method.

This analysis defines low contraceptive confidence for women who reported using a traditional method (22% of women use either withdrawal or periodic abstinence), moderate contraceptive confidence for those using a modern reversible method (eg, pill, condom and/or an intrauterine device, constituting 36%) and high contraceptive confidence for women using a permanent method (5%), either female or male. About 37% of women in the analysis sample have reported not using any method: contraceptive confidence for these women cannot be observed. We retain these women in the analysis, however, since their abortion history is still important in a context where abortion is normative fertility control behaviour. We include two controls relevant to contraceptive behaviour: the month and year of first method use and another variable measuring the previous method discontinued.

To capture the latent effect of abortion propensity, the analysis uses abortion history as a proxy measure. Unfortunately, the MDHS has not collected any data on abortion attitudes. We therefore use the proportion of pregnancies a woman has terminated. A simple count is inadequate since older women have greater exposure to multiple abortions, which may introduce bias. Using the proportion of pregnancies aborted overcomes this problem. Other than recall problems inherent in cross-sectional surveys, any deliberate under-reporting of abortion in postsocialist countries is very low.[22 28] Contraception and abortion are often seen as complementary in the Moldovan context—women report that the use of ineffective methods (such as withdrawal) combined with frequent recourse to abortion is a normative fertility control technique especially for traditional method users. An interaction between contraceptive method and abortion propensity is used to test the differential effect of abortion on different levels of contraceptive confidence.

Another key predictor variable is the marriage cohort, which is intended to capture the changes in first birth rate, which is often determined by economic circumstances, especially the availability of housing.[23 27 29] The age range of women in the data set (15–49) means that there should be some caution when interpreting results for the oldest marriage cohort since there will be some left censoring: this marriage cohort is specified, covering a wider range than others to ensure a sufficient sample size. The model controls for other effects which could potentially influence the decision to have a first birth, the ability of women to conceive and sociodemographic

characteristics. These include: age at marriage, level of education of women, geographical region and place of residence. As with the key explanatory variables, some of these are proxy variables limited to information available at survey. For example, the duration of the first marriage is used to estimate whether the woman was in a continuous marital union prior to first birth and whether union dissolution or separation occurred before the first birth. Other control variables were considered in the model as they were thought to be relevant a priori (ethnicity, wealth index, religious affiliation, employment type, seasonality of employment and receipt of family planning media messages), but were found not to significantly improve the model fit. Statistical significance was assessed by the use of the likelihood ratio test with significance at the 5% level. The model was estimated in SPSS 19.0.

## RESULTS

The regression results adjusting for relevant confounders and control variables are presented for three selected effects: (1) marriage cohorts, (2) contraceptive confidence and (3) abortion propensity. The final model is presented in online supplementary table A1. Owing to the interaction terms and time dependency specified in the model, it is difficult to interpret coefficients directly, in particular the assessment of statistical significance of overall probabilities. We therefore use this model to generate estimated survival curves and cumulative hazards, and report the cumulative hazard of

first birth at 12, 24 and 36 months after marriage as a summary statistic in table 1 as well as cumulative survival curves for each main variable examined. In the tables, to allow the reader to assess significant effects, we present CIs adjusted for pairwise comparisons at the 5% level: the non-overlap of these intervals can be interpreted as a difference which is significant at the 5% level.

## Marriage cohorts

The adjusted hazard rate of a first birth for each duration since marriage is estimated for different marriage cohorts. The results are shown in the form of survival plots (figure 1), truncated at 36 months for visual clarity. The survival plot indicates the proportion of women yet to have first birth at month t following marriage. We also report the cumulative hazard of first birth at 12, 24 and 36 months after marriage as a summary statistic in table 1A.

Women married during 1970–1979, 1980–1984 and 1985–1989 exhibited homogeneous survival trajectories, indicating rapid transition to motherhood: more specifically, 70% of women have had their first child within the first 2 years of their marriage. However, there is a distinct slowing trend in the transition to first birth within the first 24 months following marriage among those married during and after the post-independence period (1990–1994 birth cohort onwards). This trend is roughly linear as depicted in the survival curves shifting upwards, suggesting an increasing delay in first birth. The curve for the 1995–2000 cohort overlaps with the most recent cohort after 24 months, which suggests a

**Table 1** Proportion of women having had first birth 12, 24 and 26 months after marriage. All controls (age at marriage, education, residence, region, union dissolution and contraceptive uptake) are set to sample means

| | Months after marriage | | |
| --- | --- | --- | --- |
| | **12** | **24** | **36** |
| (A) Estimated adjusted cumulative hazard of first birth following marriage by marriage cohort (CIs adjusted for pairwise test of difference in proportions at 5% level) | | | |
| Marriage cohort | | | |
| 1970–1979 | 0.42 (0.41 to 0.43) | 0.77 (0.76 to 0.78) | 0.88 (0.87 to 0.89) |
| 1980–1984 | 0.40 (0.39 to 0.41) | 0.75 (0.74 to 0.76) | 0.86 (0.85 to 0.87) |
| 1985–1989 | 0.42 (0.41 to 0.43) | 0.75 (0.74 to 0.76) | 0.87 (0.86 to 0.88) |
| 1990–1994 | 0.41 (0.40 to 0.42) | 0.74 (0.73 to 0.75) | 0.83 (0.82 to 0.84) |
| 1995–1999 | 0.37 (0.36 to 0.38) | 0.66 (0.65 to 0.67) | 0.79 (0.78 to 0.79) |
| 2000 or recent | 0.32 (0.31 to 0.33) | 0.65 (0.64 to 0.66) | 0.84 (0.83 to 0.85) |
| (B) Estimated adjusted cumulative hazard of first birth following marriage by level of contraceptive confidence (CIs adjusted for pairwise test of difference in proportions at 5% level) | | | |
| Contraceptive confidence | | | |
| Low confidence | 0.43 (0.42 to 0.44) | 0.75 (0.74 to 0.75) | 0.85 (0.84 to 0.85) |
| Moderate confidence | 0.44 (0.43 to 0.45) | 0.77 (0.76 to 0.78) | 0.88 (0.87 to 0.89) |
| High confidence | 0.41 (0.40 to 0.42) | 0.74 (0.73 to 0.76) | 0.88 (0.87 to 0.89) |
| Unobserved | 0.33 (0.32 to 0.34) | 0.64 (0.63 to 0.65) | 0.74 (0.73 to 0.74) |
| Abortion propensity | | | |
| No propensity | 0.41 (0.39 to 0.42) | 0.73 (0.72 to 0.74) | 0.84 (0.83 to 0.85) |
| Low propensity | 0.48 (0.46 to 0.49) | 0.80 (0.79 to 0.80) | 0.89 (0.88 to 0.90) |
| Moderate propensity | 0.44 (0.43 to 0.45) | 0.76 (0.75 to 0.77) | 0.84 (0.83 to 0.85) |
| High propensity | 0.42 (0.41 to 0.43) | 0.72 (0.71 to 0.73) | 0.82 (0.81 to 0.82) |

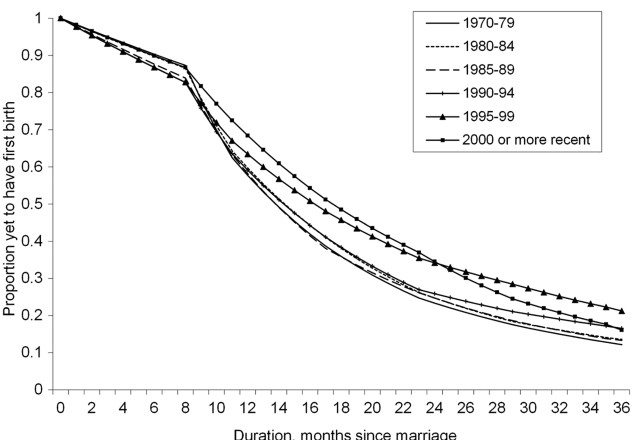

**Figure 1** Estimated survival curves by marriage cohort. Proportion of women yet to have first birth (y-axis) for months postmarriage (x-axis) based on predictions from full model. Curves are disaggregated by marriage cohort. All controls (type of contraceptive method, abortion propensity, age at marriage, education, residence, region, union dissolution and contraceptive uptake) are set to sample means.

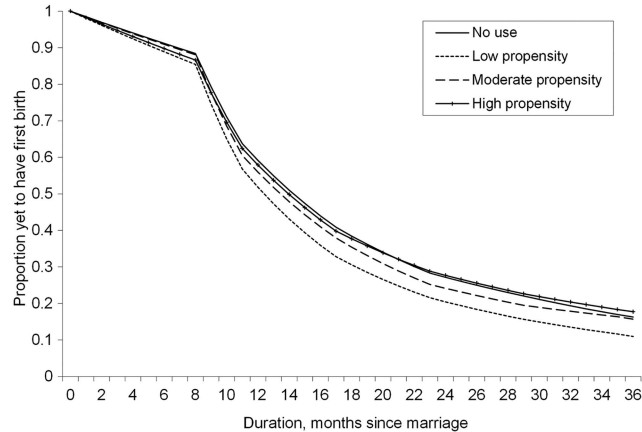

**Figure 2** Estimated survival curves by level of contraceptive confidence at mean abortion propensity. Proportion of women yet to have first birth (y-axis) for months postmarriage (x-axis) based on predictions from full model. Curves are disaggregated by contraceptive confidence. All controls (abortion propensity, age at marriage, education, residence, region, union dissolution and contraceptive uptake) are set to sample means.

propensity for early transition to motherhood among recently married women. That said, the overall probability of having a birth remains relatively constant—for instance 3 years following marriage the 1990–94, 1995–99 and 2000 or more recent cohorts have the same cumulative hazard of birth as the presocialist (1970–79, 1980–84 and 1985–89) marriage cohorts. This is largely due to the recuperation effect 2–3 years following marriage, suggesting that although the interval between marriage and first birth is longer, the probability of giving a birth does not vary across cohorts.

This is also reflected in the cumulative hazard, with the hazard among the preindependence cohorts at 41%, 75% and 86% for 12, 24 and 36 months, respectively. However, there is a considerable fall in the cumulative hazard for the 1995–1999 and 2000 marriage cohorts, indicating the increasing delay of first birth following the collapse of Socialism, but overall Moldovan women have a consistently high probability of becoming mothers.

### Contraceptive confidence

The estimated survival curve for each level of contraceptive confidence is presented in figure 2. Cumulative hazards are presented in table 1B. Owing to the interaction between contraceptive confidence and abortion propensity, these estimated survival plots are generated where the categories of abortion propensity are set to their sample proportions. All other covariates are held constant, producing net effects controlling for selected characteristics controlling for marriage cohort effects and socioeconomic characteristics.

Among women with a measurable contraceptive level (ie, where a contraceptive method is recorded at survey), the survival curve for high contraceptive confidence is the

highest, indicating the slowest transition to first birth in this group. The first birth rate is higher for women with moderate contraceptive confidence, compared with women with low contraceptive confidence. The survival curve for high contraceptive confidence is comparable to those of the low confidence group until 24 months following marriage (indeed, there is no statistically significant difference detectable at this point), when there is a rapid fall in the proportion of women yet to have first births. This indicates that, in general, low contraceptive confidence is associated with low hazard of first birth and hence longer duration between marriage and first birth. On the other hand, an increase in contraceptive confidence is associated with increased hazard of first birth, which clearly suggests rapid transition to motherhood among women with high confidence.

### Abortion

The estimated survival curve of first birth for women with low contraceptive confidence is presented in figure 3, which examines the association between low contraceptive confidence and abortion propensity. In general, the proportion of women yet to have first birth is high for women with no abortion propensity, and the survival curves are lower for women with low and moderate abortion propensity. Table 1B presents the estimated cumulative hazard of first birth. Broadly, we see that the probability of having first birth is low for women with no abortion propensity. However, the cumulative hazard of first birth is significantly higher at 12, 24 and 36 months among low abortion users and 12 and 24 months among moderate abortion users following marriage. This suggests that overall women who were prepared to use abortion at least partially have a shorter interval between

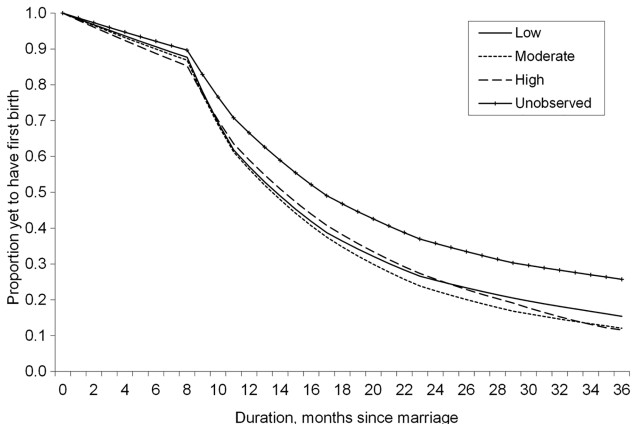

**Figure 3** Estimated survival curve by propensity to use abortion among women traditional method users. Proportion of women yet to have first birth (y-axis) for months postmarriage (x-axis) based on predictions from full model. Curves are disaggregated by abortion propensity. All controls (age at marriage, education, residence, region, union dissolution and contraceptive uptake) are set to sample means. Contraceptive confidence is set to low.

marriage and first birth. The survival curve for women with high abortion propensity is roughly comparable or slightly lower than women with no abortion propensity. We cannot detect an effect for high abortion prevalence. Indeed, there is some evidence of attenuation in the higher cumulative hazard of first birth at higher abortion levels: at 12, 24 and 36 months, the cumulative hazard is lower for moderate and high abortion users than for women with low abortion propensity.

## CONCLUSION

This study examined the impact of contraceptive confidence on the shifts in timing of first birth in a low fertility regime with high abortion rates. The analysis yielded three key findings. First, there is evidence of contraceptive confidence effect on the timing of first birth: women with low contraceptive confidence tend to delay their first birth, while women with high contraceptive confidence progress more rapidly to motherhood. The results supported the hypothesis that women using effective methods have increased contraceptive confidence and a relatively shorter interval between marriage and first birth than users of ineffective methods. This result has wide-ranging implications in the low fertility context of Moldova where modern methods are not widely available and many women rely on traditional methods for fertility control. Second, overall use of abortion results in a shorter interval between marriage and first birth, particularly for women with low contraceptive confidence. We do note, however, that this effect is non-linear: an increasing propensity to use abortion (eg, high compared with low propensity) will tend to depress overall fertility behaviour. Abortion appears to be an effective substitute for women with low contraceptive confidence, suggesting that voluntary

abortion tends to potentially outweigh traditional method failure. An efficient strategy to reduce increasing abortion rates, therefore, is to increase access to modern methods to young couples in Moldova. Third, the study provides evidence of increase in the duration between marriage and first birth for recent marriage cohorts, although motherhood is still common among Moldovan women. This development is consistent with the increasing trend in fertility postponement behaviour as well as increasingly complex co-relationships between fertility and marriage in the Moldovan setting,[23] reflecting that the increased heterogeneity and complexity of union-fertility interactions is typical of broader westernisation and modernisation processes underway in Moldova[23] and perhaps wider changes characterising the second demographic transition.[17] This is also partly explained by the economic changes in postsocialist Europe and increasing aspirations of women to establish a career before childbearing.[27]

**Contributors** ML-A, SSP and GBD made substantial contributions to the conception, analysis and interpretation of data; drafting of the article and its critical revision for important intellectual content; and gave final approval of the version to be published.

**Funding** The UK Economic and Social Research Council (Ref: PTA-031–2006-00188).

**Competing interests** None.

**Ethics approval** University of Southampton.

**Provenance and peer review** Not commissioned; externally peer reviewed.

**Data sharing statement** No additional data are available.

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
