## [Reviewer comments · BMJ Open]

Some articles will have been accepted based in part or entirely on reviews undertaken for other BMJ Group journals. These will be reproduced where possible.

ARTICLE DETAILS

TITLE (PROVISIONAL)	Contraceptive Confidence and Timing of First Birth in Moldova: An event history analysis of retrospective data
AUTHORS	Lyons-Amos, Mark; Padmadas, sabu; Durrant, Gabriele

VERSION 1 - REVIEW

REVIEWER	Krystof ZEMAN Wittgenstein Centre for Demography and Global Human Capital (IIASA, VID/ÖAW, WU), Vienna, Austria
REVIEW RETURNED	26-Mar-2014

GENERAL COMMENTS	In my understanding the presented results of statistical analysis do not agree with the conclusions of the paper. First key finding of the paper is: "there is evidence of contraceptive confidence effect on the timing of first birth: women with low contraceptive confidence tend to delay their first birth, while women with high contraceptive confidence progress more rapidly to motherhood." However, the lines at figure 2 look very similar, with low and high contraceptive-confident slightly delaying compared to moderate-confident. In table 1 this relationship shows no statistical significance. Elsewhere in the paper it is also stated that: "low contraceptive confidence is associated with a low hazard of a first birth and hence longer duration between marriage and first birth. On the other hand, an increase in contraceptive confidence is associated with an increased hazard of a first birth, which clearly suggests rapid transition to motherhood among women with high confidence." But in the data the low contraceptive confidence (i.e. traditional method users) shows rather high relative hazard ratio of first birth, with no statistical significance against modern methods. Second key finding of the paper is: "greater use of abortion results in shorter interval between marriage and first birth particularly for women with a low contraceptive confidence." But figure 3 shows that greater use of abortion results rather in longer interval (and lower, not higher, cumulative hazard of first birth, as stated in section 3.3). Either I completely do not understand the argumentation or it must be wrong. I was a bit afraid about spurious effect of premarital conception; cannot it be that women of stronger religiosity, who refuse modern contraception and abortion, have sex only after marriage, while among non-religious women there is much higher proportion of premarital conception? Did you try to control for religion? Also, why don't you show figures 1-3 right from the beginning at marriage and
--

	proportion 1? Education have quite strong effect, can it be that it interacts with contraception and abortion use, with the first birth interval, and with premarital conceptions?
--	---

REVIEWER	Jirina Kocourkova Charles University in Prague, Faculty of Science, Department of Demography and Geodemography
REVIEW RETURNED	04-Apr-2014

GENERAL COMMENTS	The paper addresses the issue of recent changes in timing of first birth in Moldova. These changes are explained by contraceptive confidence hypothesis, i.e. using effective contraceptive methods results in shortening interval between marriage and first birth. However, timing of first birth can be influenced by many important factors of external (economic and housing situation) and personal character (health status, partner relationship), etc. How were these factors controlled for in the study? If not it should be specified as limitations. Moreover, most of Eastern European countries have been facing profound changes in family formation and reproductive behaviour called second demographic transition: childbearing postponement, decrease in marriage, increase in proportion of children born out of wedlock, etc. That seems to go against research question. Using modern contraception enables women to plan childbearing more effectively but rather postpone it even within marriage. How was it reflected in the study? If Moldova is an exceptional case, important demographics regarding fertility and marriage (trends in TFR, in marriage rate, in mean age of mother at first birth or proportion of children born out of wedlock) should be provided for Moldova and results cannot be taken as general. Post socialist Eastern Europe has significantly differentiated in using birth control methods in the last two decades. Some countries (Czech Republic or Slovakia) have experienced profound decline in abortion rate and rapid increase in use of effective contraception methods. Limited access to modern method is not valid any more in most of Eastern European countries. Most of the references, particularly in Introductory part, are not up-to-date. While examining first birth can be taken as an extension of the contraceptive confidence hypothesis based on some relevant references, hypothesis regarding abortion use is not convincing as it is not adequately explained or based on relevant references. Moreover, discussion regarding results is not appropriately developed.
--

VERSION 1 – AUTHOR RESPONSE

Reviewer Name Krystof ZEMAN

Institution and Country Wittgenstein Centre for Demography and Global Human Capital (IIASA, VID/ÖAW, WU), Vienna, Austria

Please state any competing interests or state 'None declared': None

In my understanding the presented results of statistical analysis do not agree with the conclusions of the paper.

First key finding of the paper is: "there is evidence of contraceptive confidence effect on the timing of first birth: women with low contraceptive confidence tend to delay their first birth, while women with

high contraceptive confidence progress more rapidly to motherhood."

However, the lines at figure 2 look very similar, with low and high contraceptive-confident slightly delaying compared to moderate-confident. In table 1 this relationship shows no statistical significance.

Elsewhere in the paper it is also stated that: "low contraceptive confidence is associated with a low hazard of a first birth and hence longer duration between marriage and first birth. On the other hand, an increase in contraceptive confidence is associated with an increased hazard of a first birth, which clearly suggests rapid transition to motherhood among women with high confidence."

But in the data the low contraceptive confidence (i.e. traditional method users) shows rather high relative hazard ratio of first birth, with no statistical significance against modern methods.

Thank you for this comment. We believe our argument and interpretation to be correct, and have included a number of amendments to make this clear. Firstly, while the reviewer is correct in that certain elements of the main effect or certain parts of interaction terms may not demonstrate statistical significance, the table is only included for completeness and intellectual openness, and is difficult to interpret directly (we highlight this on pp. 8, ln 16). For this reason, we included the figures 1-3 for an overall trend and tables 1-3 for specific interpretation. To be able to interpret statistical significance, we have edited tables 1-3 to include confidence intervals adjusted for pairwise comparisons (so that low contraceptive confidence can be compared to medium, low can be compared to high etc.). This is highlighted at pp. 8 ln 20. Where we are able to detect statistically significant difference, these are consistent with our interpretation (i.e. cumulative hazards are higher for methods with higher levels of contraceptive confidence).

Second key finding of the paper is: "greater use of abortion results in shorter interval between marriage and first birth particularly for women with a low contraceptive confidence."

But figure 3 shows that greater use of abortion results rather in longer interval (and lower, not higher, cumulative hazard of first birth, as stated in section 3.3).

The reviewer has pointed out a nuance in the results that we had previously over simplified. Overall, the non-use of abortion tends to have the lowest cumulative hazard of first birth: both low and moderate abortion use have significantly higher hazards of first birth- consistent with our previous interpretation. However, this effect is non-linear: increasing abortion propensity tend to attenuate this effect to the point that there is no significant elevation in cumulative hazard for high propensity. We now note this in our interpretation (pp. 11) and conclusion (pp. 12 ln 14).

I was a bit afraid about spurious effect of premarital conception; cannot it be that women of stronger religiosity, who refuse modern contraception and abortion, have sex only after marriage, while among non-religious women there is much higher proportion of premarital conception? Did you try to control for religion?

This was one of the control variables that we tried but was not significant. We have noted this at pp. 8 ln. 6

Also, why don't you show figures 1-3 right from the beginning at marriage and proportion 1?

This was due to the low hazard in the 9 months following marriage, which we felt initially were not instructive. In light of the question raised, we have redrawn the graphs to include this phase.

Education have quite strong effect, can it be that it interacts with contraception and abortion use, with the first birth interval, and with premarital conceptions?

We have included education as a main effect control. We are concerned that extending the model to

include further interactions may start to introduce structural problems in estimation due to the already large number of interactions for contraceptive method and abortion use.

Reviewer Name Jirina Kocourkova

Institution and Country Charles University in Prague, Faculty of Science, Department of Demography and Geodemography

Please state any competing interests or state 'None declared': None declared

The paper addresses the issue of recent changes in timing of first birth in Moldova. These changes are explained by contraceptive confidence hypothesis, i.e. using effective contraceptive methods results in shortening interval between marriage and first birth. However, timing of first birth can be influenced by many important factors of external (economic and housing situation) and personal character (health status, partner relationship), etc. How were these factors controlled for in the study? If not it should be specified as limitations.

The reviewer is correct, union formation and first birth behaviours are one of the most complicated demographic processes. While we tested the significance of all the control variables available, and where possible attempt to control for relevant factors (a full list is now included at pp8 ln 6), the reviewer is right to sound a note of caution, and as such we have included the caveat that our model relates to only one aspect of first birth behaviour (please see pp 2 ln 11).

Moreover, most of Eastern European countries have been facing profound changes in family formation and reproductive behaviour called second demographic transition: childbearing postponement, decrease in marriage, increase in proportion of children born out of wedlock, etc. That seems to go against research question. Using modern contraception enables women to plan childbearing more effectively but rather postpone it even within marriage. How was it reflected in the study?

Thank you for this comment. The reviewer is correct in the sense that, as we noted in the original manuscript, the second demographic transition is a potential explanation for cross cohort- differences. However, potential explanations for the changes seen in fertility and partnership behaviour in Eastern Europe are myriad, and there is no universally accepted explanation for their cause (pp. 2 ln. 5). Moreover not all of the trends of the second demographic transition are present in Moldova (we note this at pp 2 ln 4.). As such, seeking to include an explanation of the cause in fertility behaviour is beyond the scope of this paper. We have however included a note to the effect that due to influences such as the SDT, it is important to disaggregate results by cohort (see pp 2 ln 8), the potential explanatory role the second demographic transition plays (e.g pp 3 ln20) and acknowledge the importance of the SDT in the conclusion (pp 12 ln 25).

If Moldova is an exceptional case, important demographics regarding fertility and marriage (trends in TFR, in marriage rate, in mean age of mother at first birth or proportion of children born out of wedlock) should be provided for Moldova and results cannot be taken as general. Post socialist Eastern Europe has significantly differentiated in using birth control methods in the last two decades. Some countries (Czech Republic or Slovakia) have experienced profound decline in abortion rate and rapid increase in use of effective contraception methods.

We now note that Moldovan fertility has not recovered in the same way that other countries in the region have (pp. 2 ln 1.) and have clarified when we refer to Moldova (for example pp1 ln7) . Moreover, we note that the share of traditional methods in Moldova that much of Eastern Europe at pp 2. ln 18.

Limited access to modern method is not valid any more in most of Eastern European countries. We have changed relevant references to refer more specifically to Moldova (pp 1 In 7).

Most of the references, particularly in Introductory part, are not up-to-date. The contraceptive confidence hypothesis is an under researched topic and existing literature sparse as we highlight on pp 1 In 16. We have included an additional, more recent, reference (reference 11) pertaining to the contraceptive confidence hypothesis.

VERSION 2 – REVIEW

REVIEWER	Krystof ZEMAN Wittgenstein Centre for Demography and Global Human Capital (IIASA, VID/ÖAW, WU), Vienna, Austria
REVIEW RETURNED	06-Jun-2014

GENERAL COMMENTS	There are some missing or misspelled words, especially in the new added parts:  -page 2, line 1...missing "TFR" -p.2,l.19...former-socialist, not Soviet -p.3,l.21 -p.11,l.13 -p.11,l.19...36, not 26 months -I could not find footnotes A,B,C -some new parts of page 2 could be better formulated -please reformulate new sentence on p.11,l.13-15 (...at least partially...), with connection to the following two sentences I accept the paper only with reservation, as I feel the second key finding of the paper is quite inconclusive, without strong statistical evidence. However this fact is now addressed in the new version of the paper.
---

REVIEWER	Jirina Kocourkova Charles University in Prague, Faculty of Science, Department of Demography and Geodemography
REVIEW RETURNED	06-Jun-2014

GENERAL COMMENTS	My previous comments were taken into account adequately.
--

VERSION 2 – AUTHOR RESPONSE

We have taken on board to comments of reviewer 1 in particular, and have made the appropriate typographical and spelling corrections. We hope that the paper now meets the required level of precision. Since the amendments were minor, I have included these changes only in the marked copy- the response to reviewers from the previous round are retained in the submission for reference.